# Transcriptome Analysis Reveals the Molecular Regularity Mechanism Underlying Stem Bulblet Formation in Oriental Lily ‘Siberia’; Functional Characterization of the *LoLOB18* Gene

**DOI:** 10.3390/ijms232315246

**Published:** 2022-12-03

**Authors:** Shaozhong Fang, Chenglong Yang, Muhammad Moaaz Ali, Mi Lin, Shengnan Tian, Lijuan Zhang, Faxing Chen, Zhimin Lin

**Affiliations:** 1Institute of Biotechnology, Fujian Academy of Agricultural Sciences, Fuzhou 350003, China; 2College of Horticulture, Fujian Agriculture and Forestry University, Fuzhou 350002, China

**Keywords:** Lilium, sucrose, starch, cytokinin, virus-induced gene silencing (VIGS), gibberellin

## Abstract

The formation of underground stem bulblets in lilies is a complex biological process which is key in their micropropagation. Generally, it involves a stem-to-bulblet transition; however, the underlying mechanism remains elusive. It is important to understand the regulatory mechanism of bulblet formation for the reproductive efficiency of Lilium. In this study, we investigated the regulatory mechanism of underground stem bulblet formation under different conditions regarding the gravity point angle of the stem, i.e., vertical (control), horizontal, and slanting. The horizontal and slanting group displayed better formation of bulblets in terms of quality and quantity compared with the control group. A transcriptome analysis revealed that sucrose and starch were key energy sources for bulblet formation, auxin and cytokinin likely promoted bulblet formation, and gibberellin inhibited bulblet formation. Based on transcriptome analysis, we identified the *LoLOB18* gene, a homolog to *AtLOB18*, which has been proven to be related to embryogenic development. We established the stem bud growth tissue culture system of Lilium and silenced the *LoLOb18* gene using the VIGS system. The results showed that the bulblet induction was reduced with down-regulation of *LoLOb18*, indicating the involvement of *LoLOb18* in stem bulblet formation in lilies. Our research lays a solid foundation for further molecular studies on stem bulblet formation of lilies.

## 1. Introduction

Lilies (*Lilium* spp.), a group of monocotyledonous ornamental plants, are widely grown for commercial purposes. They contribute significantly to the global ornamental flower industry. They are commonly marketed as outdoor and indoor fresh-cut flowers and potted plants, and for landscaping in private gardens [1,2]. Lilies can be propagated through both vegetative and sexual propagation. The main advantages of using bulblets for commercial production are rapid cloning, genetic purity, and seasonal supply [3]. In past years, scientists focused on the in vitro formation of lily bulblets and, in this respect, great progress occurred [4,5,6]. However, the mechanism of bulblet formation in natural lilies remains unknown. The bulblets are formed in the leaf axils of the underground stem and are common in many cultivars. Meanwhile, bulbils are generated exclusively from the leaf axils in the middle-upper portion of the aboveground stem, which is a rare phenomenon in Lilium species [7]. In the oriental lily ‘Siberia’, bulblets which form underground are usually collected for commercial production [8]. Their quality and production quantity and can be improved by cultivation measures [9].

Phytohormones (i.e., auxins, gibberellins, cytokinin, and abscisic acid) have been proven to be involved in bulblet formation and the regulation of plant growth and development under stress conditions [10,11,12]. Exogenous treatment of 6-BA significantly induces bulbil formation in the aboveground stem of LA hybrid lily. In addition, underground stem bulblets were also formed earlier [12]. Higher auxins, gibberellins, and jasmonic acid levels, and low levels of abscisic acid might facilitate bulblet formation in scale cuttings [13]. The content of endogenous gibberellin regulates new bulblet formation in *L. lancifolium* [14]. Although dynamic changes of different phytohormones have exhibited distinct regulation patterns in bulblet formation, numerous studies have found that auxin and cytokinin are significant for stem bulblet formation. Auxins likely promote the initiation of bulblet formation and then inhibit their further growth [7], while cytokinin promotes regulated bulblet formation [15]. Considering the already known phenomenon of the gravity-induced asymmetric distribution of phytohormones [16], especially auxin [17] and cytokinin [18], we set different gravitropic point angles in underground stem by cultivation measures and revealed the relationship between gravity angle and bulblet formation.

Several studies have revealed that genetic factors are critical for stem bulblet formation in lilies [7]. Type-B response regulators (*LIRRs*) are involved in bulbil formation in a functionally redundant manner and can activate *LIRR9* expression [19]. The *LlWOX9* and *LlWOX11* genes have been shown to be positive regulators of bulbil formation. Additionally, type-B *LIRRs* may bind to the promoters of *LlWOX9* and *LlWOX11* and promote their transcription [20]. In LA hybrid lily, *LaKNOX1* was shown to interact with *LaKNOX2* and *LaBEL1* to regulate stem bulblet formation [12]. In other species, *KNOX* genes have also been associated with bulbil formation, like *AtqKNOX1* and *AtqKNOX2* in *Agave tequilana* [21]. Strigolactones (SLs) related gene *CCD8* also has a similar function and is associated with bulbil formation in lily [22] and *solanum tuberosum* [23]. According to the aforementioned studies, several genes related to bulblet formation have similar functions in different species. Therefore, it seems evident that the mechanism of bulblet formation in different species is conservative. The plant-specific *LOB* (lateral organ boundaries domain) gene family plays a fundamental role in the regulation of the lateral development of plant organs [24]. In Arabidopsis, *AtLOB16*, *AtLOB18,* and *AtLOB29* combinatorically regulate lateral root organogenesis as direct targets of Aux/IAA-ARF modules in the Auxin signaling pathway [25,26,27]. The LOB genes *AtLOB16*, *AtLOB18*, and *AtLOB29* have also been identified as key regulators of callus induction in various organs [28]. However, the role of *LOB18* in bulblet formation of lilies is still unclear.

In this study, we revealed the regulation of stem bulblet formation by changing the gravitropic point angle in the underground stem. The role of starch and sucrose metabolism and plant hormone signal transduction in the bulblet formation of lilies was identified using transcriptome analysis. To further understand the molecular mechanism of bulblet formation, we identified and characterized an LBD gene *LoLOB18*, a homolog of *AtLOB18*. Laying the bulb horizontally on the seed bed could endow lilies with a strong self-propagation ability. This measure can be applied to lily production to accelerate reproduction. As well as providing comprehensive transcript information, this work identified the *LoLOB18* gene function in stem bulblet formation, laying a solid foundation for further molecular studies on stem bulblet formation.

## 2. Results

### 2.1. Phenotypic Analysis

To investigate the effect of different gravitropic point angle on bulblet formation, lily bulbs were divided into three groups when planting (Figure 1A). As shown in Figure 1B, horizontally placed bulbs showed more curvy stems than others (20 days after planting) and no bulbils formed. Furthermore, the gravitropic point angle in the underground stems affected the number and quality of new bulblets. We found that the average numbers and weight of bulblets in horizontally placed bulbs were higher than in the vertical and slant groups. Additionally, the slant group also performed better than the control group in terms of the quantity and quality of new bulblets (Figure 1D).

### 2.2. Transcriptome Sequencing and Transcript Assembly

The cDNA libraries from twenty samples produced 122.92 Gb clean reads, i.e., ≥6.4 Gb per sample. The error rate for clean reads was 0.03% (Appendix A). In this study, a de novo assembly strategy was adopted due to the lack of reference genome sequences for Lilium. In total, 145,530 unigenes were assembled with a mean length of 819 bp (N50 length of 1071 bp). These results indicate that the quality of transcriptome sequencing and assembly in this study was reliable.

### 2.3. Functional Annotation and Classification of Unigenes

In order to determine the gene function of unigenes, 145,530 unigenes were annotated based on seven databases (Nr, Nt, Pfam, KOG, Swiss-Prot, KEGG, GO). In total 63,762, 53,444, 27,326, 56,595, 54,137, 54,133, and 24,441 unigenes were annotated, respectively (Appendix A). Approximately 43.81% of the unigenes were matched to the NR database. The result indicated that more than half of the sequences failed to match homologs. Therefore, the transcriptomes of Lilium will contribute to the study of genes with novel functions.

With a screening threshold of *p*adj < 0.5 and the fold change in the Nr annotation database, a total of 17,935 DEGs were identified based on a comparison of three groups, i.e., the control group (I) vs. the horizontal group (II), the slant group (III) vs. the control group (I), and the horizontal (II) vs. the slant group (III), during two different development stages (A and B) in Lilium. As shown in Figure 2A, there were 2750 DEGs between AⅡ vs. AⅠ, 2062 DEGs between AⅢ vs. AⅠ, 9599 DEGs between BⅡ vs. BⅠ, and 9607 DEGs between BⅢ vs. BⅠ.

We selected 17,935 DEGs to predict functions by GO annotation. The results of the GO assessment were divided into biological process, cellular component, and molecular function categories (Figure 2B). Obviously, in the molecular function term, DEGs were significantly enriched in terms of catalytic and transferase activity. According to the KEGG results (Figure 2C), DEGs were mapped to the predicted metabolic pathway and significant enrichment was determined with a screening threshold of *p*adj < 0.5. The annotated changes between the three comparison groups were found to be involved in plant hormone signal transduction and starch and sucrose metabolism.

### 2.4. Analysis of DEGs and Candidate Gene Selection

At stage Ⅰ (20 days after planting), most of the differentially expressed genes related to starch and sucrose metabolism pathways showed stable expression levels (Figure 3). However, at stage Ⅱ (50 days after planting), most of the genes showed significantly higher expression levels in horizontal and slant group than vertical (control) group, such as sucrose synthase (*SUS*) (cluster-98963.26735, cluster-98963.38015, cluster-87536.0 and cluster-98963.75221), hexokinase (*HK*) (cluster-65351, cluster-56215 and cluster-98963.22366), UTP-glucose-1-phosphate uridylyltransferase *(UGP*) (cluster-98963.21368), ADPG pyrophosphorylase (*AGPS*) (cluster-989863.8332, cluster-95130 and cluster-91520), soluble starch synthase (*SSS*) (cluster-88881), and starch branching enzyme (*SBE*) (cluster-25037). These results suggest that starch biosynthesis probably provides energy for stem bulblet initiation.

The KEGG enrichment analysis shown in Figure 2C indicated that a large number of DEGs were assigned to plant hormone signal transduction pathways. Most of the DEGs were enriched in the auxin pathway and exhibited higher expression levels in the horizontal and slant groups during both sampling stages (Figure 4). Several genes that catalyzed the biosynthesis of abscisic acid (ABA) were up-regulated in the horizontal and slant groups during stage Ⅱ, like protein phosphatase 2C (*PP2C*) (cluster-98963.13777, cluster-98963.17123, and cluster-98963.10655)*,* serine/threonine-protein kinase (*SNRK2*) (cluster-98963.9163), and ABA responsive element binding factor (*ABF*) (cluster-98963.33646, cluster-46254 and cluster-94852.1). In the brassinosteroid (BR), cytokinin (CK), and jasmonic acid (JA) pathways, most DEGs were up-regulated in the horizontal and slant groups during bulblet formation. In addition, in GA signal transduction, the expression of F-box protein (*GID2*) (cluster-98963.12179 and cluster-98963.26958), DELLA protein (*RGL1*) (cluster-67540), DELLA protein (*SLR1*) (cluster-98963.24377), and phytochrome-interacting factor 4 (*PIF4*) (cluster-98963.9362) were obviously down-regulated. Those results revealed that GA signal transduction could inhibit stem bulblet formation.

In order to further study the mechanism of stem bulblet formation and the selection of key genes, we ranked the DEGs during bulb formation based on the absolute value of log^2^ fold change. We selected the DEGs top 20 ranking for further analysis. The 13 differentially expressed genes ranked highest in both the horizontal and slant groups during stage Ⅱ (Table 1). Among them, cluster-39434 is homologous to *AtLOB18*. In a previous study, *AtLOB18* was identified as key regulator of plant regeneration in *A. thaliana* [29]. As such, *LoLOB18* probably plays a crucial role in stem formation.

### 2.5. Validation of Transcriptome Data with qRT-PCR Analysis

To ensure the reliability of RNA-Seq, seven unigene-related plant hormones signaling transduction and starch and sucrose metabolism were randomly selected from the identified DEGs for qRT-PCR validation. A comparative analysis of all the selected genes showed a similar expression pattern in qRT-PCR to that observed in RNA-Seq data (Figure 5), suggesting the reliability of the results.

### 2.6. Candidate Gene Cloning and Characterization Analysis

We cloned the sequence of *LoLOB18* based on transcriptome data (Appendix A). The *LoLOB18* encoded the protein sequences of 229 amino acids. A peptide analysis showed that *LoLOB18* contained one typical LOB domain. By blasting in the Smart database, it was found that the sequence of *LoLOB18* had a similar LOB domain to other related species (Figure 6A). Furthermore, the phylogenetic analysis was performed on the *LoLOB18* and LOB gene family members of *A. thaliana* (Figure 6B). The results revealed that this novel LOB gene in lily is clustered closely to *A. thaliana LOB18 (AtLOB18).*

### 2.7. LoLOB18 Silencing Inhibits Stem Bulblet Formation

To further understand whether *LsLOB18* is involved in bulblet formation, we carried out VIGS experiments. The TRV2-LOB18 silencing vector was constructed by designing a specific primer for the *LOB18* gene (Appendix A). The coat protein in pTRV2 and the insert fragment in pTRV2 were used for detection. The results shown in Appendix A indicate that TRV2 and TRV2-LOB18 were successfully inserted and expressed in the genome of Lilium. The q-PCR was generally used for validity checks. Our experiment results show that after silencing *LOB18*, the expression of the corresponding silenced gene was decreased (Figure 7B), as was the rate of stem bulblet induction (Figure 7C). These results indicate that *LoLOB18* is closely associated with stem bulblet formation in lilies (Figure 7A).

## 3. Discussion

### 3.1. Different Hormones Regulate Bulblet Formation

The regeneration of plant organs is controlled by internal and external factors, i.e., the source and physiological state of the explants and the type and concentration of hormones used in culture [30,31,32,33,34]. Multiple hormones have been shown to regulate Lilium bulblet formation in vitro [35]. The formation of aerial bulbils originates from the axillary meristem [15,36,37]. Previous studies suggested that auxin and cytokinin play important roles in axillary meristem initiation [38]. Auxin promotes bulblet initiation and inhibits the further formation of Lilium [7]. In our study, a large number of *GH3* genes reducing IAA concentrations were up-regulated in two comparison groups, i.e., BⅡ vs. BⅠ and BⅢ vs. BⅠ. This result further verified that a decrease in auxin concentration is beneficial for stem bulblet formation. Studies also revealed that in maize [39], *Arabidopsis* [40], and potato [41], auxin inhibits the outgrowth of axillary organs. Numerous studies have proven that cytokinin is involved in bulbil formation. Exogenous cytokinin treatment could significantly improve the rate of bulbil induction in *Dioscorea Zingiberensis* [42] and *Solanum tuberosum* [43]. In this study, a KEGG enrichment analysis showed that genes related to cytokinin were up-regulated in BⅡ vs. BⅠ and BⅢ vs. BⅠ, implying that cytokinin promotes stem bulblet formation. BR-, ABA-, ethylene-, and JA-related genes also showed the same expression patterns.

### 3.2. Starch Synthesis Contributes to Bulblet Formation

The initiation of new plant organs depends on starch accumulation for energy requirements [44,45,46,47]. To date, studies on sucrose metabolism have indicated that starch and sucrose are crucial for the formation and development of bulblets [48,49]. In recent years, substantial efforts have centered on the starch regulatory enzyme ADP-glucose pyro-phosphorylase (*AGPase*) due to its pivotal role in starch biosynthesis [50,51,52,53]. In our study, two out of the three *AGPase* genes (cluster-98963.8332, cluster-95130) exhibited higher expression in the slant and horizontal groups. The over-expression of *AGPase* in maize [54], rice [55], and wheat [56] results in both an increase in seed number and enhanced vegetative growth. Thus, the up-regulation of *AGPase* genes may contribute to bulblet formation. Other sucrose and metabolism-related genes, such as *INV*, *SUS*, *UGP*, *SSS,* and *SBE,* all showed up-regulated expression in the slant and horizontal groups, indicating that sucrose and starch are crucial for bulblet formation.

### 3.3. LoLOB18 Is Positive Regulator during Underground Bulblet Formation

The *LBD* gene family is involved in many biological processes during the growth and metabolism of higher plants, such as meristem differentiation [28], leaf polarity [57], inflorescence formation [58], lateral root formation [25], callus formation [59], anthocyanin metabolism [60], and nitrogen metabolism [61]. In *Arabidopsis*, *LOB16*, *LOB19,* and *LOB18* have been reported to be involved in the control of lateral root formation [26,27]. *Plt3*, *plt5,* and *plt7* have lost the ability to regenerate adventitious bulbs due to defects in root primordium development [62]. In addition, the regeneration system of adventitious buds induced by root primordia has been reported, indicating that such buds originate from cells with root primordia properties [63,64]. In our study, silencing *LoLOB18* reduced the rate of stem bulblet induction, indicating that *LoLOB18* contributes to stem bulblet formation in Lilium.

Gravity regulates the transport and local accumulation of auxin [12]; thus, auxin acts as a crucial signal for lateral organ formation by inducing the expression of the *LOB* gene family [25,26,27]. In *Arabidopsis*, *LOB* and *KNOX* genes may interact with each other antagonistically to control lateral organ development [19]. Additionally, *KNOX* genes have been associated with organogenesis during bulblet formation, and it has been proven that *LaKNOX1* interacts with *LaKNOX2* and *LaBEL1* to regulate stem bulblet formation in lily [16]. Our transcriptome analysis revealed that the highest number of DEGs was involved in IAA signaling transduction during bulblet formation when changes in angle of gravity occurred and the expression of *LoLOB18* changed. We further found that silencing *LoLOB18* could enhance the expression of *LoKNOX1* (Figure 8 and Appendix A).

## 4. Materials and Methods

### 4.1. Plant Material and Samples Collection

The oriental hybrid species Cv. Siberia was used in this study. Three different planting patterns were adopted, i.e., planting the bulb vertically (control), laying the bulb horizontally, and placing the bulb in a slanting position on the seed bed (Figure 1A). Every group contained 20 plants. The growth medium contained 70% peat moss and 30% perlite (pH 5.5). All bulbs were propagated in plastic pots approximately 15 cm in diameter after germination. Experimental plants were grown in a greenhouse at the Institute of Biotechnology, Fujian Academy of Science, Fuzhou, China, under temperature conditions of 25 °C during the day and 20 °C at night. Plants were watered as needed and kept moist.

According to a previous study [11], the bulblets formed in the reproductive stage and can easily be seen 50 days after planting. Samples were collected at two growth stages, i.e., vegetative growth, when plants attained 20 cm height, i.e., approximately 20 days after planting (Figure 1B), and the reproductive growth stage, when the lilies expanded their leaves and underground bulblets started growing, i.e., 50 days after planting (Figure 1C). As there were no underground stem bulblets in the vegetative growth stage, all samples were collected from the middle of the underground stem node. At the vegetative growth stage, the samples from the vertical, horizontal, and slant groups were remarked as AⅠ, AⅡ, and AIII, respectively. At reproductive growth stage, there were abundant stem bulblets in the horizontal and slant groups. All samples were collected from the middle of the stem nodes with bulblets. At the reproductive stage, samples from the vertical, horizontal, and slant groups were remarked as BⅠ, BⅡ, and BIII, respectively. Three biological replicates of each group were used for RNA extraction and transcriptome sequencing. At reproductive stage, 20 plants were randomly selected to count the number and weight of new bulblets per plant (Figure 1D). All samples were frozen immediately in liquid nitrogen and then stored at –80 °C for further study.

### 4.2. RNA Extraction and cDNA Synthesis

The total RNA was extracted and purified using an RNAprep Pure Plant Kit (Polysaccharides and Polyphenolics-Rich, Tiangen, Beijing, China) [65]. RNA degradation and contamination were monitored on 1% agarose gels. RNA purity was checked using a NanoDrop N-1000 spectrophotometer (NanoDrop Technologies, Wilmington, DE, USA) [66]. RNA integrity was assessed using a RNA Nano 6000 Assay Kit of the Agilent Bioanalyzer 2100 system (Agilent Technologies, CA, USA). First-strand cDNA was synthesized from 1 µg of total RNA using the Prime Script RT Reagent Kit with a gDNA Eraser (TaKaRa, Dalian, China).

### 4.3. Library Construction and Transcriptome Sequencing

Sequencing libraries were generated using the NEBNext^®^ Ultra™ RNA Library Prep Kit from Illumina^®^ (NEB, Ipswich, MA, USA), following the manufacturer’s recommendations. Index codes were added to attribute sequences to each sample. The library generation involved five steps, as follows: firstly, the mRNA was purified and fragmented; secondly, the double-stranded cDNA was synthesized using the fragmented mRNA; thirdly, the sticky ends of the short fragments were repaired with end repair reagents to avoid self-connection; fourth, the sequencing adaptors were added to the cDNA fragments that were then enriched by PCR amplification; and finally, the quality control of the constructed libraries was assessed on the Agilent Bioanalyzer 2100 system. The clustering of the index-coded samples was performed on a cBot Cluster Generation System using TruSeq PE Cluster Kit v3-cBot-HS (Illumia, Ipswich, MA, USA) according to the manufacturer’s instructions. After cluster generation, the library preparations were sequenced on an Illumina Hiseq platform and paired-end reads were generated [46,67].

### 4.4. RNA-Sequencing Data Analysis

In order to ensure the accuracy and reliability of the RNA-sequencing data, some poor-quality reads were eliminated from the raw reads. Transcriptome assembly was accomplished using Trinity (http://www.trinity-software.com/ (accessed on 22 January 2021)) with the default parameters. Absolute values of the log^2^ (fold change) ≥ 1 and *p*-value < 0.05 were set as the threshold for significantly differential expression. Functional annotation information for DEGs was obtained by using the following databases: NR, NT, KO, Swissport, PFAM, GO, and KOG.

### 4.5. RNA Extraction and qRT-PCR

Total RNA from the middle of underground stem node was extracted at different growth stages of lily with an RNAprep Pure Plant Kit (TIANGEN, Beijing, China), according to the kit protocol. The cDNA synthesis strand was performed by using a Rever Aid First Strand cDNA Synthesis Kit (Thermo Fisher, Waltham, MA, USA), according to the manufacture’s instructions. Gene-specific primers for qRT-PCR were designed with Primer 6.0 (Appendix A). Taq Pro Universal SYBR qPCR Master Mix (Vazyme Bioech, Nanjing, China) was used in the reaction mixture according to the manufacture’s instruction. All qRT-PCRs were performed using the QuanStudio TM Real-Time PCR system under the following conditions: 95 °C for 3 min, followed by 40 cycles of 95 °C for 15 s, 56 °C for 15 s, and 72 °C for 15s. Melting curves were recorded after the 40th cycle by increasing the temperature stepwise by 0.5 °C every 5 s from 65 °C to 95 °C. The *EF-1a* gene was used as the internal control for normalization [68]. The relative gene expression levels were determined using 2^−ΔΔCT^ approach [69].

### 4.6. Isolation of LoLOB18 Gene

According to our transcriptome data (accession number: PRJNA763773), the specific primers were designed using Primer 6 (https://www.primer-e.com/ (accessed on 14 June 2021)) to clone the *LoLOB18* gene. The sequences of the primers used for PCR amplification are shown in Appendix A. Conserved protein domains were analyzed using SMART (http://smart.embl.de/ (accessed on 15 March 2021)). A phylogenetic analysis was performed using MEGA-6 (https://www.megasoftware.net/ (accessed on 15 March 2021)). Multiple sequence alignments were analyzed using the DNAMAN (v 10) software package.

### 4.7. Virus-Induced Gene Silencing (VIGS)

For *LoLOB18* silencing, the specific coding region of *LoLOB18* gene was used for VIGS vector construction. The gene-specific fragment of ~300bp was inserted into the pTRV2 vector at *Eco*RⅠ side and *Xho*Ⅰ side using a cloneExpress^®^ MultiS One Step Cloning Kit (Vazyme Biotech, Nanjing, China). The primer pairs used to generate the TRV vector are shown in Appendix A. The restriction sites are shown in Appendix A. The resulting vectors TRV2::*LOB18*, TRV2::empty, and TRV1 were transformed into *Agrobacterium tumefaciens* (strain GV3101). *A. tumefaciens* cells harboring TRV1 and TRV2::*LOB18*, TRV2 (as a negative control) were resuspended in induction medium at 1:1 ratio, OD_600_ = 1. Stem segments of Lilium were surface sterilized with 75% alcohol and 10% NaClO and then cut into small stem segments for vacuum infiltration [19]. The infiltrated segments were washed with distilled water three times for 3 min each time and then grown on MS medium with 30 g/L sucrose and 6 g/L agar, PH 5.8, in the dark at 15 °C for 2 days, followed by growth at 22 °C under a 16/8 h light/dark cycle. The rate of stem bulblet formation was assessed after two weeks of culture. The RNA of the infiltrated stem segments was extracted to measure the expression of the target genes. The relative gene expression levels were determined using the 2^−ΔΔCT^ approach [69]. The coat protein was used for positive detection. The primer pairs used for VIGS experiment are shown in Appendix A. Each treatment consisted of three experimental replicates, with 30 stem segments per replicate.

### 4.8. Statistical Analysis

All data are presented as the mean ± standard deviation (SD) of at least three independent replicates. Duncan’s multiple range test at *p* < 0.05 or *p* < 0.01 was performed with the SPSS (version 17.0, USA) statistical package.

## 5. Conclusions

In this study, the quality and quantity of Lilium bulblets were investigated by changing the gravity point angle of the underground stem. A transcriptome analysis revealed that sucrose and starch were crucial energy sources for bulblet formation. Furthermore, auxin and cytokinin were found to promote bulblet formation, while gibberellins inhibited it. In addition, the *LoLOB18* gene was identified among the DEGs, and further, it was proved by VIGS experiment that *LoLOB18* was a positive regulator during bulblet formation. However, the mechanism by which *LoLOB18*, as a transcription factor, affects bulblet formation by regulating downstream target genes is unclear. Our research laid a solid foundation for further molecular studies on the bulblet formation of Lilium.

## Figures and Tables

**Figure 1 ijms-23-15246-f001:**
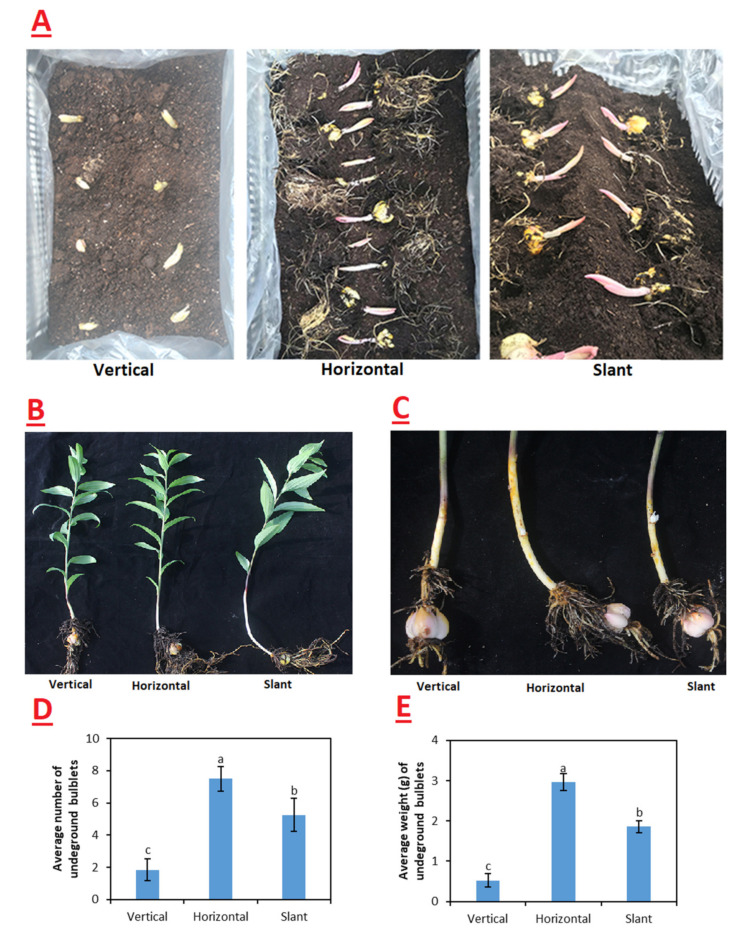
(**A**) Three different bulb planting patterns adopted in the study. (**B**) The growth of whole plants at stage A (20 days after planting). (**C**) The growth of underground stems at stage B (50 days after planting). (**D**) The average number of underground bulblets (20 plants per group). (**E**) Average weight of underground bulblets (20 plants per group). Lowercase letters indicate significance among treatment groups according to Duncan’s multiple range test at *p* < 0.05.

**Figure 2 ijms-23-15246-f002:**
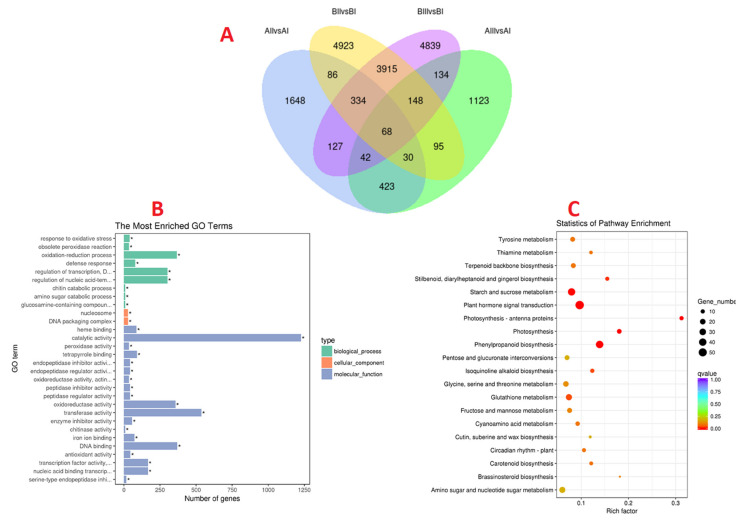
(**A**) Venn diagram of DEGs among the four comparisons performed (AⅡ vs. AⅠ, AⅢ vs. AⅠ, BⅡ vs. BⅠ and BⅢ vs. BⅠ). AI—vertical (control) group after 20 days of planting; AII—horizontal group after 20 days of planting; AIII—slant group after 20 days of planting; BI—vertical (control) group after 50 days of planting; BII—horizontal group after 50 days of planting; BIII—slant group after 50 days of planting. (**B**) GO (Gene ontology) classification histogram of the DEGs. “*” represents significant enrichment (*p* < 0.05). (**C**) Bubble chart of the DEGs enrichment pathway.

**Figure 3 ijms-23-15246-f003:**
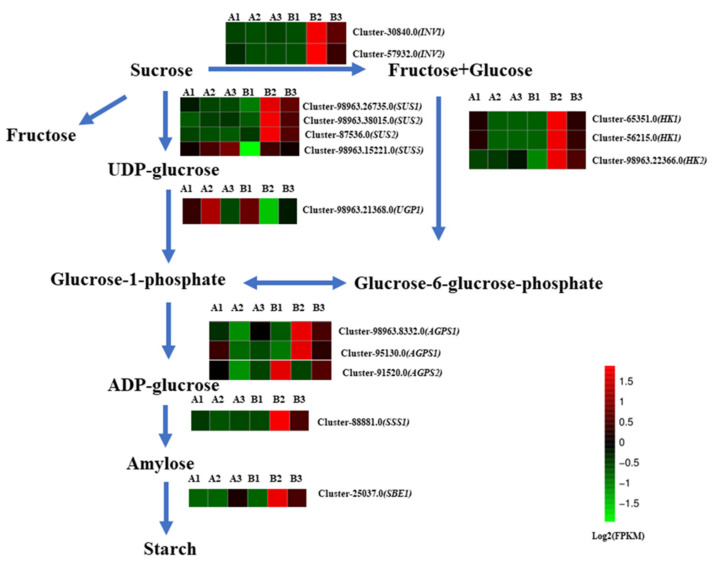
Expression patterns of the expressed genes assigned to starch biosynthesis in oriental lily. A1—vertical (control) group after 20 days of planting; A2—horizontal group after 20 days of planting; A3—slant group after 20 days of planting; B1—vertical (control) group after 50 days of planting; B2—horizontal group after 50 days of planting; B3—slant group after 50 days of planting. The log-transformed expression values range from −1.5 to 1.5. Red and green colors indicate up- and down- regulated transcripts, respectively.

**Figure 4 ijms-23-15246-f004:**
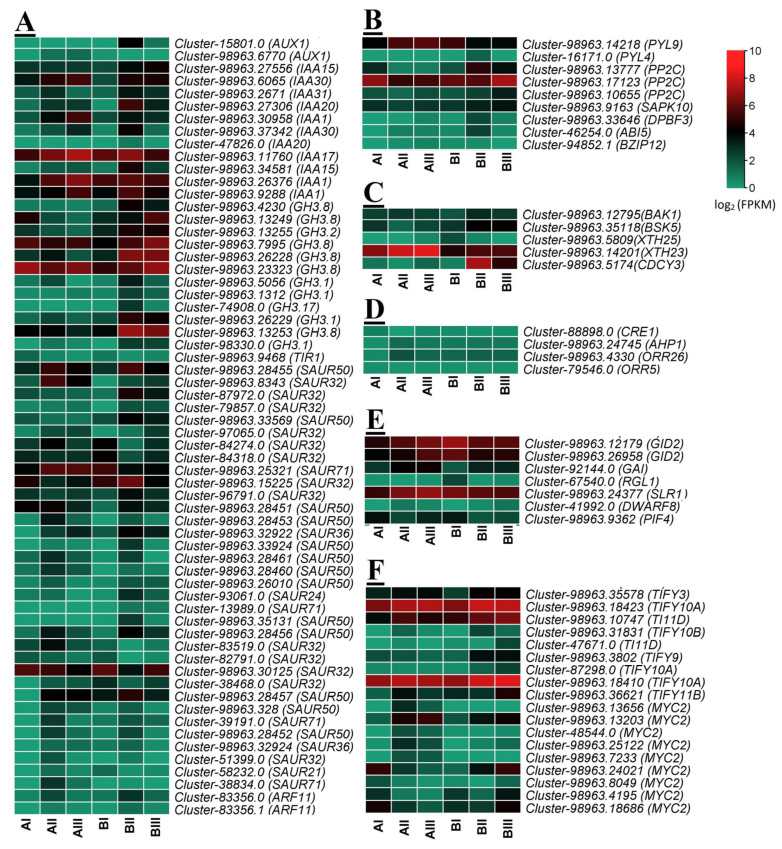
Expression patterns (FPKM) of the expressed genes in the six transcriptomes, assigned to auxin (**A**), abscisic acid (**B**), brassinosteroids (**C**), cytokinin (**D**), gibberellins (**E**), and jasmonic acid (**F**) biosynthesis, respectively. Red and green colors indicate up- and down- regulated transcripts, respectively. AI—vertical (control) group after 20 days of planting; AII—horizontal group after 20 days of planting; AIII—slant group after 20 days of planting; BI—vertical (control) group after 50 days of planting; BII—horizontal group after 50 days of planting; BIII—slant group after 50 days of planting.

**Figure 5 ijms-23-15246-f005:**
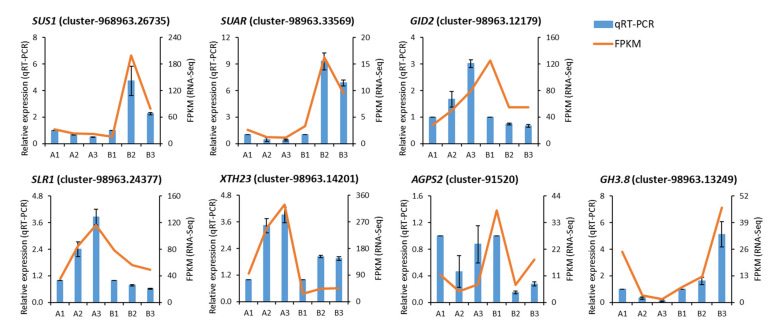
Validation of the selected DEGs by qRT-PCR analysis. A1—vertical (control) group after 20 days of planting; A2—horizontal group after 20 days of planting; A3—slant group after 20 days of planting; B1—vertical (control) group after 50 days of planting; B2—horizontal group after 50 days of planting; B3—slant group after 50 days of planting.

**Figure 6 ijms-23-15246-f006:**
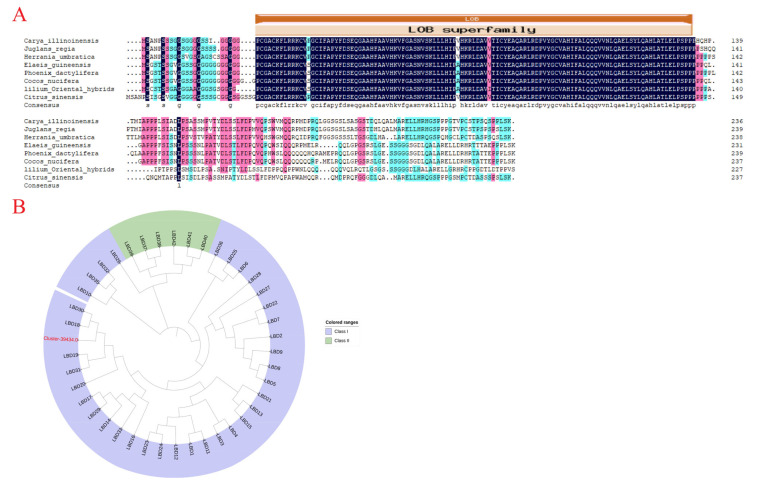
(**A**) Alignment of deduced amino acid sequences of LoLOB18 with *Carya illinoinesis* CiLOB18, *Juglans regia* JrLOB18, *Herrania umbratica* HuLOB18, *Elaeis guineensis* EgLOB18, *Phoenix dactylifera* PdLOB18, *Cocos nucifera* CnLOB18, and *Citrus sinensis* CsLOB18. LOB domain composes the red box. Identical amino acid residues are shaded in dark blue, similar in pink, and less similar in light bule. (**B**) A phylogenetic tree analysis of LoLOB18 with *LOB* family genes from *Arabidopsis thaliana*. The maximum likelihood (ML) method was used for construction.

**Figure 7 ijms-23-15246-f007:**
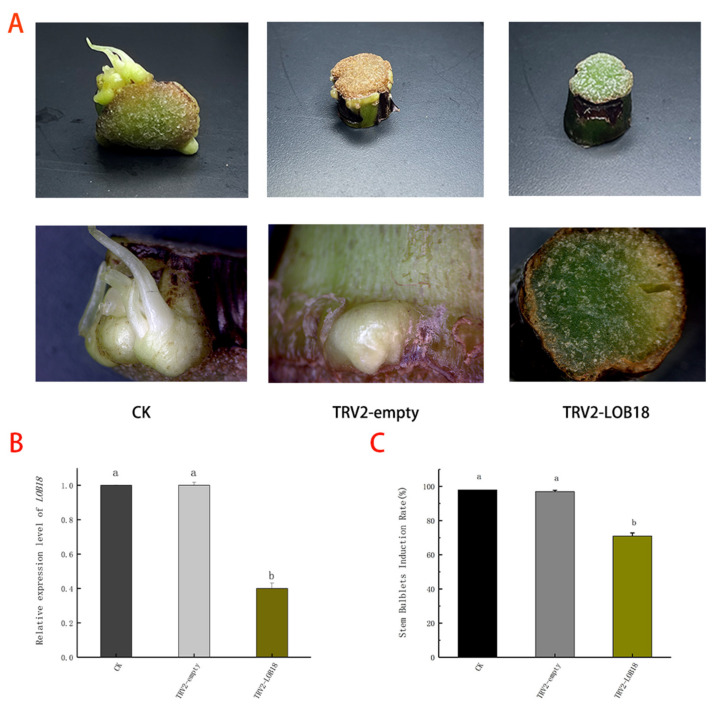
(**A**) The phenotype of stem segments after VIGS treatment. (**B**) Gene expression after the silencing of the *LoLOB18*. (**C**) The rate of stem bulblet formation after two weeks of culture. Values are means ± SDs (n = 3). Lowercase letters (a–c) indicate statistically significant differences at *p* < 0.05.

**Figure 8 ijms-23-15246-f008:**
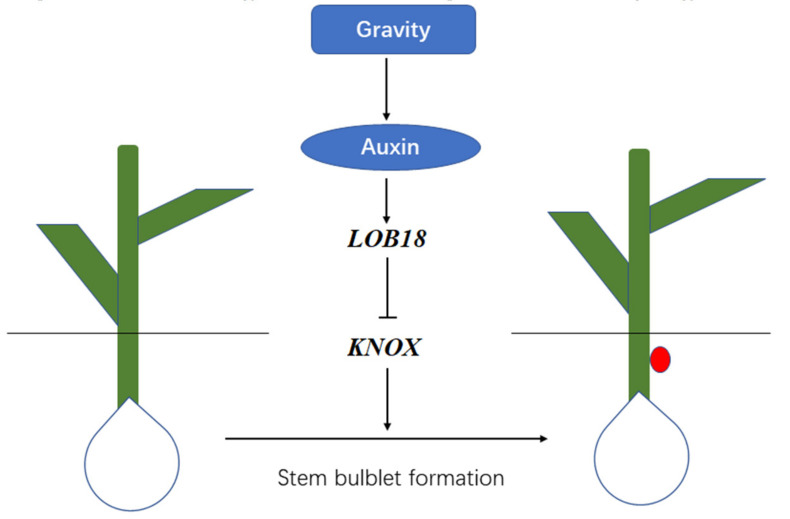
The putative regulatory mechanism of *LoLOB18* gene cooperation with auxin signaling under variation of gravity to regulate the underground stem bulblet formation in lily. “↓” indicates promoted activity and “⊥” indicates inhibited activity.

**Table 1 ijms-23-15246-t001:** Candidate genes selected for further study of bulblet formation mechanism based on absolute value of log^2^ fold change.

ID	Homologous Gene/Protein	Regulate Pattern (BIII vs. BI)	Regulate Pattern (BII vs. BI)	Log^2^ FC	
				BIII vs. BI	BII vs. BI
Cluster-98963.5096	SWEET14	UP	UP	25.294	14.737
Cluster-98963.36497	SWEET14	UP	UP	13.14	11.302
Cluster-95907.0	HSR201	DOWN	DOWN	−12.161	−12.058
Cluster-98963.11855	SLC50A	UP	UP	12.024	14.801
Cluster-98963.32942	SWEET14	UP	UP	12.001	14.567
Cluster-94100.0	CYP89A2	DOWN	DOWN	−11.983	−11.879
Cluster-98963.3555	E2 binding domain	UP	UP	11.954	10.721
Cluster-98963.31619	GAUT12S	UP	UP	11.589	11.511
Cluster-98963.30325	PER64	UP	UP	10.841	11.787
Cluster-39434.0	LOB18	UP	UP	10.258	11.872
Cluster-98963.3508	RSI1	UP	UP	10.732	12.466
Cluster-98156.0	non-specific lipid-transfer protein	UP	UP	10.667	11.265
Cluster-89656.0	PBL28	UP	UP	10.59	10.543

## Data Availability

The datasets presented in this study can be found in online repositories. The names of the repository/repositories and accession number(s) can be found below: https://www.ncbi.nlm.nih.gov/, PRJNA763773.

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
