# Peer review of "Transcriptome Analysis Reveals the Molecular Regularity Mechanism Underlying Stem Bulblet Formation in Oriental Lily ‘Siberia’; Functional Characterization of the LoLOB18 Gene"

_ijms, 2022, doi:10.3390/ijms232315246_

Round 1
Reviewer 1 Report
This manuscript provides an important study on the regulation of stem bulblets' formation in lily, a phenomenon which has not yet been fully elucidated. The manuscript is in general very clearly written and presented. However, there is a lack of precision in the description of the samples used for the study. Which bulblets were exactly sampled? What were the stages? Although there is information about this in a previous publication, it's important to better describe this in the present manuscript to enable the reader to relate the results to specific tissues and developmental stages.
Please see also remarks in the attached file.

Author Response
This manuscript provides an important study on the regulation of stem bulblets' formation in lily, a phenomenon which has not yet been fully elucidated. The manuscript is in general very clearly written and presented. However, there is a lack of precision in the description of the samples used for the study. Which bulblets were exactly sampled? What were the stages? Although there is information about this in a previous publication, it's important to better describe this in the present manuscript to enable the reader to relate the results to specific tissues and developmental stages.
Response: Thank you so much for the suggestion. To describe clearly the details of the samples used for the study, we have made appropriate corrections. In lines 302 – 310, we described more details of samples used for RNA extraction and transcriptome sequencing.
Please see also remarks in the attached file.
Comments in the attached file:
Table 1 should be in supplementary data.
Response: Table 1 has been moved to supplementary data (now Table S1).
Was any treatment given to the bulbs before planting?
Response: There was no special treatment given to the bulbs before planting.
Please give more precisions about the samples and their collection
Response: To clearly describe the details of samples used for the study, we have made appropriate corrections in “Plant Material and Samples Collection”.
Reviewer 2 Report
The study "Transcriptome Analysis Reveals the Molecular Regularity 2 Mechanism Underlying Stem Bulblet Formation in Oriental 3 Lily ‘Siberia’; Functional Characterization of the LoLOB18 4 Gene" sounds interesting and well-designed. There are several problems, that need to address before further processing.
1: Why author select 50 days for transcriptome analysis? and how many plants were used as independent replicates?
2: Line 148-150, write the complete name of genes then used brackets for abbreviation.
3: Line 164-165, write the complete name of genes then used brackets for abbreviation.
4: Figure 4, all the heat maps have different read expressions. The author needs to bring it in a consistent format, so readers can easily find the expression difference between each hormone. secondly, the 0 indication is down-regulation? how about the genes that have - minus expression? According to my point of view, figure 4 needs to reconstruct similarly to like figure 3. So in the whole paper, we can the same idea of up and down-regulation.
5: The discussion part is very weak, the author needs to make a molecular mechanism/ pathway figure as a model figure and explain it in detail in figure legends. And conclude the molecular mechanism identified in Lily. This is very important.
6. Explain the soil medium composition used for lily cultivation.
7: Highlight the restriction site used for cloning in the supplementary file.
Author Response
The study "Transcriptome Analysis Reveals the Molecular Regularity Mechanism Underlying Stem Bulblet Formation in Oriental Lily ‘Siberia’; Functional Characterization of the LoLOB18 4 Gene" sounds interesting and well-designed. There are several problems, that need to address before further processing.
1: Why author select 50 days for transcriptome analysis? and how many plants were used as independent replicates?
Response: To clearly describe the details of samples used for the study, we have made appropriate corrections in “Plant Material and Samples Collection”. According to previous publication [1], the 50 days after planting is a critical stage for bulblet formation, and three plants from each group were randomly selected as independent replicates.
- Zhang, Y.; Yong, Y.B.; Wang, Q.; Lu, Y.M. Physiological and Molecular Changes during Lily Underground Stem Axillary Bulbils Formation. Russ. J. Plant Physiol. 2018, 65, 372–383, doi:10.1134/S1021443718030172.
2: Line 148-150, write the complete name of genes then used brackets for abbreviation.
Response: Thank you, we have made appropriate corrections in line 148-151.
3: Line 164-165, write the complete name of genes then used brackets for abbreviation.
Response: Thank you, we have made appropriate corrections in line 165-175.
4: Figure 4, all the heat maps have different read expressions. The author needs to bring it in a consistent format, so readers can easily find the expression difference between each hormone. secondly, the 0 indication is down-regulation? how about the genes that have - minus expression? According to my point of view, figure 4 needs to reconstruct similarly to like figure 3. So in the whole paper, we can the same idea of up and down-regulation.
Response: The figure 4 has been reconstructed as per suggestions.
5: The discussion part is very weak, the author needs to make a molecular mechanism/ pathway figure as a model figure and explain it in detail in figure legends. And conclude the molecular mechanism identified in Lily. This is very important.
Response: Thank you very much for the constructive comment. We have provided a putative regulatory mechanism in Figure 8. As the description in line 275- 285, we have addressed the logic between bulblet formation, plant hormone and LOB18 gene for explaining the bulblets formation affected by gravity point angle of underground stems.
- Explain the soil medium composition used for lily cultivation.
Response: The growing media contained 70% peat moss and 30% perlite (pH 5.5) (see lines 293-294). Furthermore, the growth media and other conditions for VIGS have also been explained in lines 381-384.
7: Highlight the restriction site used for cloning in the supplementary file.
Response: We added the supplementary file Figure S3 to indicate the restriction site.
Round 2
Reviewer 2 Report
The reviewer has improved the quality according to my suggestions. I am satisfied now.